# Searching through functional space reveals distributed visual, auditory, and semantic coding in the human brain

Sreejan Kumar[1,2], Cameron T. Ellis[2], Thomas P. O'Connell[3], Marvin M. Chun[2], Nicholas B. Turk-Browne[2]*

**1** Princeton Neuroscience Institute, Princeton University, Princeton, New Jersey, United States of America,
**2** Department of Psychology, Yale University, New Haven, Connecticut, United States of America,
**3** Department of Brain and Cognitive Sciences, MIT, Cambridge, Massachusetts, United States of America

* nicholas.turk-browne@yale.edu

**Data Availability Statement:** The authors confirm that all data underlying the findings are fully available without restriction. The data used in this study are available here: https://dataspace.

## Abstract

The extent to which brain functions are localized or distributed is a foundational question in neuroscience. In the human brain, common fMRI methods such as cluster correction, atlas parcellation, and anatomical searchlight are biased by design toward finding localized representations. Here we introduce the functional searchlight approach as an alternative to anatomical searchlight analysis, the most commonly used exploratory multivariate fMRI technique. Functional searchlight removes any anatomical bias by grouping voxels based only on functional similarity and ignoring anatomical proximity. We report evidence that visual and auditory features from deep neural networks and semantic features from a natural language processing model, as well as object representations, are more widely distributed across the brain than previously acknowledged and that functional searchlight can improve model-based similarity and decoding accuracy. This approach provides a new way to evaluate and constrain computational models with brain activity and pushes our understanding of human brain function further along the spectrum from strict modularity toward distributed representation.

## Author summary

There are two classical views about how the mind is organized in the brain. Early phrenology and neurophysiology and later neuropsychology argued that brain regions are specialized for certain functions of the mind. Older behavioral neuroscience and more recent neural network modeling and pattern classification instead argued against a one-to-one mapping, and rather that functions of the mind are distributed across multiple brain regions. Although there is considerable evidence for both perspectives in modern cognitive neuroscience, we hypothesize that the degree to which functions are distributed has been underestimated because of biases in prior work that favored finding specialized regions. Our novel machine learning approach, functional searchlight, reveals that features of a movie extracted with three different types of computational model and object

princeton.edu/jspui/handle/88435/
dsp01nz8062179 and https://github.com/
psychoinformatics-de/studyforrest-data-phase2.
The code for the method developed in this
publication can be found here: https://github.com/
sreejank/DistributedCodingBrain.

**Funding:** This work was supported by NSF grants
CCF 1839308 (NBTB) and BCS 1558497 (MMC),
NIH grants R01 MH069456 (NBTB) and R01
MH108591 (MMC), and the Canadian Institute for
Advanced Research (NBTB). The funders had no
role in study design, data collection and analysis,
decision to publish, or preparation of the
manuscript.

**Competing interests:** The authors have declared
that no competing interests exist.

representations are more widely distributed in the brain than suggested by current methods. Moreover, these distributed representations carry more movie content than could previously be decoded from the brain. This suggests a better way to conduct model-based analysis of brain data and provides a more solid basis on which to evaluate and refine theoretical models.

## Introduction

One of the most important debates throughout the history of neuroscience has been whether each mental function is localized to a dedicated brain region or distributed across regions [1]. Early studies of patients with specific brain damage and accompanying behavioral deficits suggested that some functions, such as language and executive control, can be localized. This localist perspective provided the foundation for initial studies of the healthy brain with functional magnetic resonance imaging (fMRI), which identified regions of interest with circumscribed functions [2]. Subsequent studies, however, showed support for a distributional perspective by suggesting that some functions instead arise out of the joint action of multiple regions [3]. Such claims were supported by the emergence of multivariate methods that decode the function of patterns of fMRI activity across populations of voxels [4].

Despite the promise of multivariate methods, the predominant exploratory approach for finding distributed representations with them remains inherently localist. Specifically, patterns of activity are extracted from small, contiguous anatomical volumes by moving a cube or sphere of voxels, known as a "searchlight", throughout the brain [5]. These patterns of activity are passed on for subsequent multivariate analysis, such as decoding the category of a stimulus. Although a valuable tool for mapping the informational contents of individual regions, information spread across disparate regions is never included within the same searchlight and thus neglected. One potential solution is to extract whole-brain patterns of activity [6]. However, even if the information is distributed throughout the brain, only a fraction of voxels would be expected to contribute, hence these multivariate models would be hard to fit and suffer from the curse of dimensionality. Even if successful, it is hard to interrogate the model to determine how the information is represented, because it is possible to observe different voxel weights for identical activity profiles [7].

We hypothesized that for many cognitive functions, information is widely distributed throughout the brain and that evidence for localized representation may partially be attributed to anatomical constraints of current methods. To test this hypothesis, we developed a new searchlight approach that is not bound by anatomy. This *functional* searchlight retains the exhaustive search of traditional anatomical searchlight while eliminating the localist assumption that only neighboring voxels contain useful information. It is also important to note that functional searchlight is doing this exhaustive search on exactly the same brain voxels and time series as anatomical searchlight, except the voxels that comprise each searchlight are different. Specifically, we re-map voxels from their original 3-D anatomical space with coordinates in $x$ (left-right), $y$ (anterior-posterior), and $z$ (inferior-superior) dimensions into a new functional space with dimensions for orthogonal latent variables that capture reliable sources of variation in brain activity irrespective of anatomy (Fig 1).

The functional space was learned through shared response modeling (SRM) [8], which finds a $k$-dimensional representation capturing the information shared across participants viewing the same stimulus. In functional space, two voxels are arranged close together if their projection weights into the shared space are similar, that is, if they load similarly on the latent

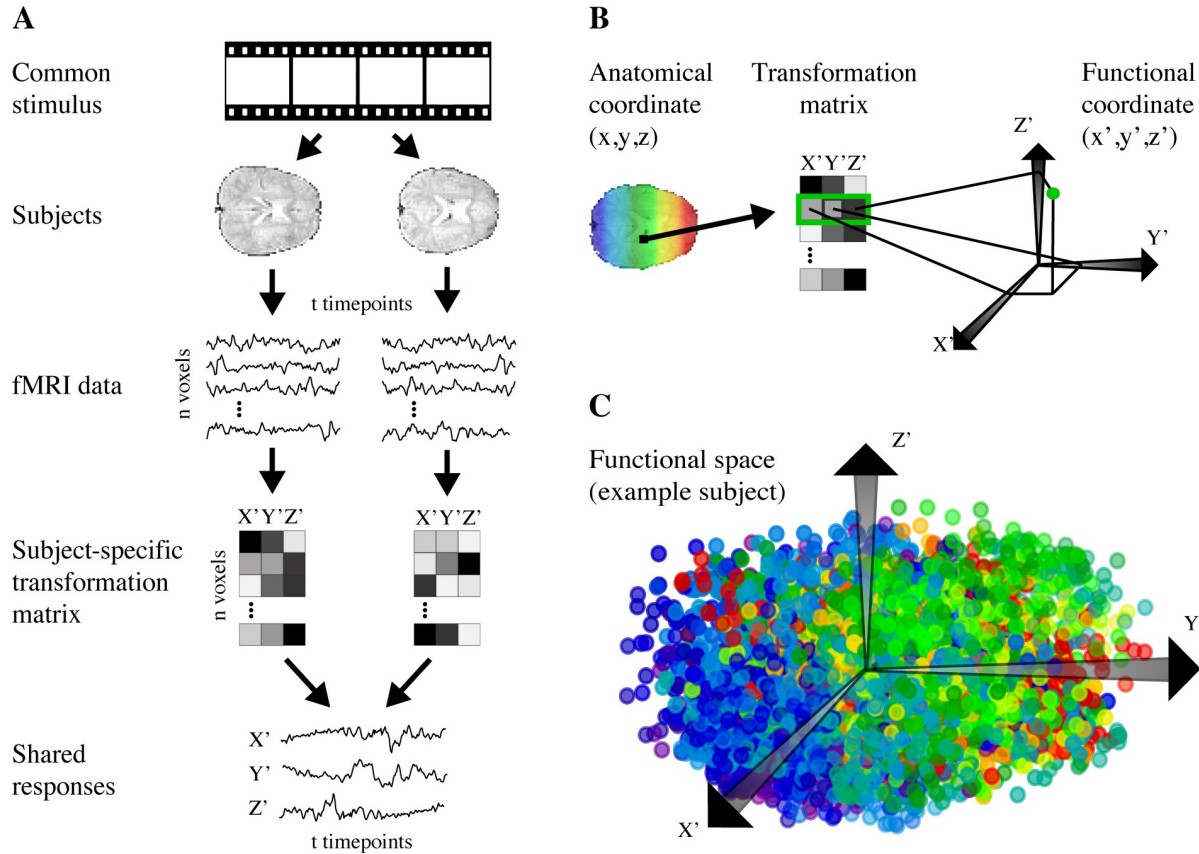

**Fig 1. Transformation from anatomical space to functional space.** (A) We use shared response modeling (SRM) to transform whole-brain data into a *k*-dimensional shared feature space. (B) Each voxel is transformed into functional space by using its loadings on the dimensions of the shared space as its coordinates in functional space. (C) Example functional space for one subject. Color indicates position on the anterior-posterior *y* axis of the input anatomical space, and as can be seen voxels get reorganized in functional space. Voxels that are functionally similar but anatomically disparate can be grouped together (e.g., blue-purple and red in top left). Note that this functional space is three-dimensional for visualization purposes, but the functional space used in our analyses had 200 dimensions.

variables. Because proximity in the functional space is governed entirely by this correspondence and not by anatomical distance, a single searchlight in this space could have access to information that is anatomically distributed throughout the brain. Just like an anatomical searchlight, all voxels in the brain will be tested still, but the functional searchlight will include different sets of voxels in the searchlights. To the extent that activity patterns extracted from these functional searchlights better represent task information than those from anatomical searchlights over contiguous voxels, the information can be said to be distributed.

This method has similarities to approaches used previously but is unique in its potential to discover distributed representations. Functional alignment has been performed within a region of interest [9] and within searchlights [10] to create denoised representations for input to MVPA. In this case, functional alignment is a preprocessing step to remove sources of noise not shared across participants and to compensate for small differences in anatomical alignment of voxels within the region of interest/searchlight. However, these methods are different from ours: we perform functional alignment on the whole brain (rather than a subset of voxels) to reorganize all voxels (rather than to denoise them) prior to running searchlight MVPA in this reorganized space. Hence, our method is uniquely positioned to reveal broadly distributed neural representations.

## Results

To first fit the SRM and compare the performance of functional vs. anatomical searchlights, we used an fMRI dataset in which participants watched an episode of BBC's "Sherlock" [11]. SRM requires a hyperparameter for the number of dimensions, which we determined to be $k = 200$ by performing time-segment matching (S1 Fig). After learning this functional space on half of the movie, we evaluated how well patterns of activity from functional searchlights encoded visual, auditory, and semantic features using model-based analysis on the other half of the movie. For each functional searchlight, the nearest 342 voxels in this high dimensional space were used to define the searchlight, hence using the same number of voxels as was used in the anatomical searchlight.

We first tested whether voxels in functional searchlights, relative to anatomical searchlights, share more information with representations in stimulus-computable models of visual and auditory processing. Deep neural networks (DNNs) offer a way to extract features from an audio-visual stimulus and measure the expression of these features in the brain [12]. We computed DNN activity separately for the video and audio components of the movie. For video, DNN activity was computed for individual frames using AlexNet [13], a DNN model pre-trained to recognize objects from natural images; hidden layer activity was averaged across video frames that fell within the same TR (1.5s). For audio, 1.5s audio segments were fed into a branching music-speech recognition DNN [14] (referred to here as KellNet).

For each layer in both the visual and auditory networks, a representational similarity matrix (RSM) was computed as the correlation matrix of the DNN activity across all time-points. For a given layer of the network, the pattern of activity across units at every time-point was correlated with every other time-point in the movie. Likewise, an fMRI RSM was computed for each searchlight as the correlation matrix of the BOLD activity patterns in the searchlight across all time-points. The upper triangular elements of these model and brain RSMs were then correlated in a second-order analysis to assess the similarity of the information captured in the fMRI searchlight and DNN layer of a given network. To minimize the contribution of the auto-correlation inherent in these data that would inflate the similarity, a buffer of 10 TRs (15s) was excluded off the diagonal and ignored for subsequent analysis. This procedure was completed for all subjects, searchlight locations, and hidden layers in both the visual and auditory DNNs.

Not all searchlights were expected to represent the audio-visual content of movie. Hence, we compared the performance of the top 1% of functional and anatomical searchlights. When we take the top 1% of each, there will be an equivalent number of searchlights in each method used for evaluation. Functional searchlights resulted in significantly higher RSA than anatomical searchlights for both visual features in AlexNet and auditory features in KellNet (for representative layers, see Fig 2A left). Indeed, functional searchlights outperformed anatomical searchlights in all AlexNet layers except the first and in all KellNet layers (Fig 3, S1 Table). The average increase in RSA collapsing across layers was 5.433% for AlexNet (95% CI = [3.709, 7.224], bootstrap $p < 0.0001$) and 8.273% for KellNet ([5.423, 11.355], $p < 0.0001$). Voxels that consistently contributed to the top 1% of functional searchlights were more distributed than those that consistently contributed to the top 1% of anatomical searchlights (Fig 2B left). The location of voxels from top-performing searchlights across individual subjects (S2 Fig) and the median Euclidean distance between these voxels (S3 Fig) indicate that the representations captured by functional searchlight were distributed and consistent across subjects. The fact that the advantage of functional searchlight over anatomical searchlight in visual and auditory analyses increased for a larger searchlight size (S4 Fig) further suggests that the voxels in functional searchlight were broadly distributed rather than locally concentrated in larger regions. By

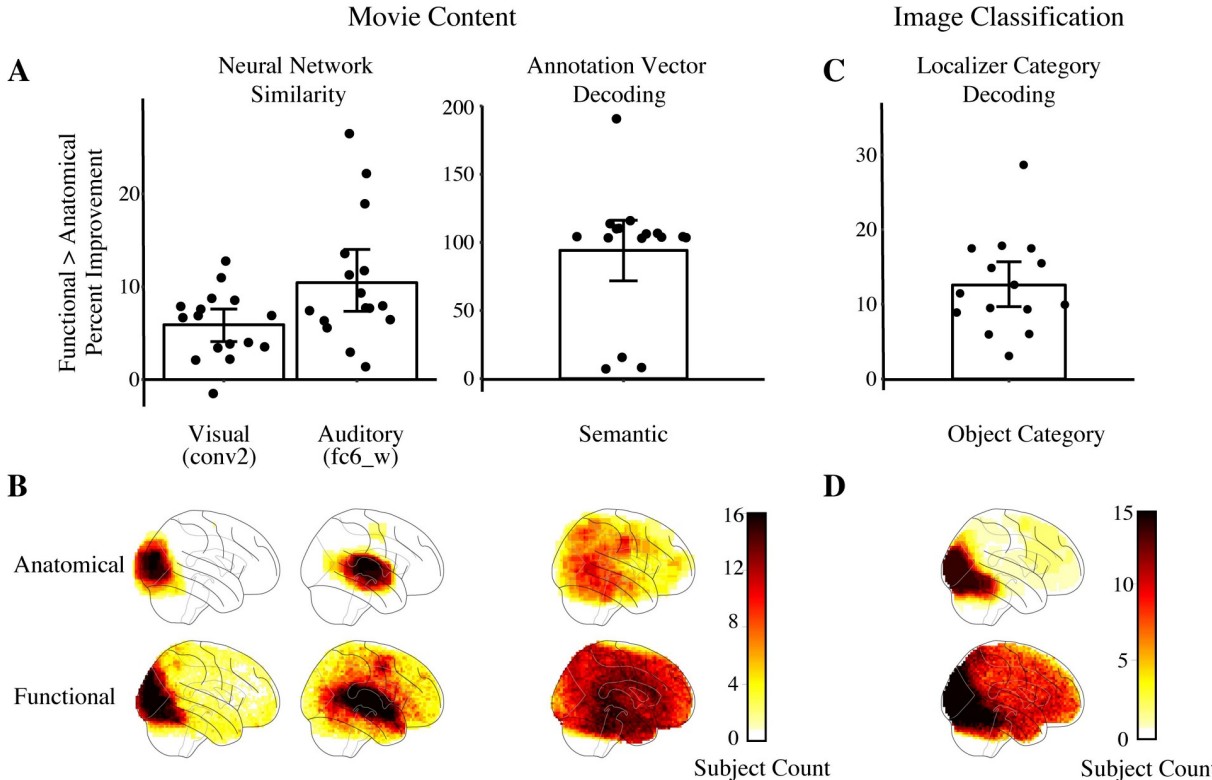

**Fig 2. Enhanced performance of multivariate analysis with functional searchlight.** We calculated the percent improvement of functional searchlight over anatomical searchlight for every subject from the top-performing 1% of searchlights of each type. (A) Each dot represents the percent improvement for a subject from an example layer in the AlexNet visual network (conv2) and the KellNet auditory network (fc7_W), as well as for annotation vector decoding. Error bars depict 95% confidence intervals (CIs) from bootstrapping. Raw performance levels for each searchlight type and non-parametric chance baselines can be found in Fig 3A. (B) For the visual and auditory analyses, we visualize which voxels contained model-based information by depicting the count of the number of subjects for whom that voxel contributed to one or more of the top 1% of their functional and anatomical searchlights. For the semantic analysis, we do the same but only visualize the center voxels of the top 1% of searchlights to avoid clutter. (C) We compared functional vs. anatomical searchlight in a localizer task by attempting to classify brain activity evoked by images from six categories: bodies, faces, houses, objects, landscapes, scrambled. Each dot represents percent improvement from chance of the mean top 1% searchlight accuracy. Error bar depicts 95% CI from bootstrapping. (D) We visualize the locations of all voxels that contributed to the top-performing searchlights for category decoding.

removing the assumption that information is anatomically local, we found neural representations that are more consistently correlated with model representations.

To generalize these findings beyond sensory systems, we then analyzed the representation of semantic content in the brain. Theories of semantic cognition [15] and recent findings [16] suggest that such content is widely distributed, and yet the extent may have been underestimated empirically with current methods. We decoded semantic content of each time-point in the movie by predicting sentence embeddings of the movie's scene annotations from brain activity [17]. We found considerably better decoding of annotation embeddings for the top 1% of functional searchlights than the top 1% anatomical searchlights (Fig 2A right, S1 Table). The average increase in annotation vector decoding was 94.155% (95% CI = [70.819, 116.024], bootstrap $p < 0.0001$). In other words, aggregating information that is anatomically distributed throughout the brain (Fig 2B right) provided a more accurate representation that was beneficial in probing neural representations of semantic content. Although the voxels from the top-performing searchlights in the semantic analysis were highly distributed across the cortex in every subject (S2 Fig), the spatial distribution of these voxels was not as consistent across subjects as the visual and auditory analyses.

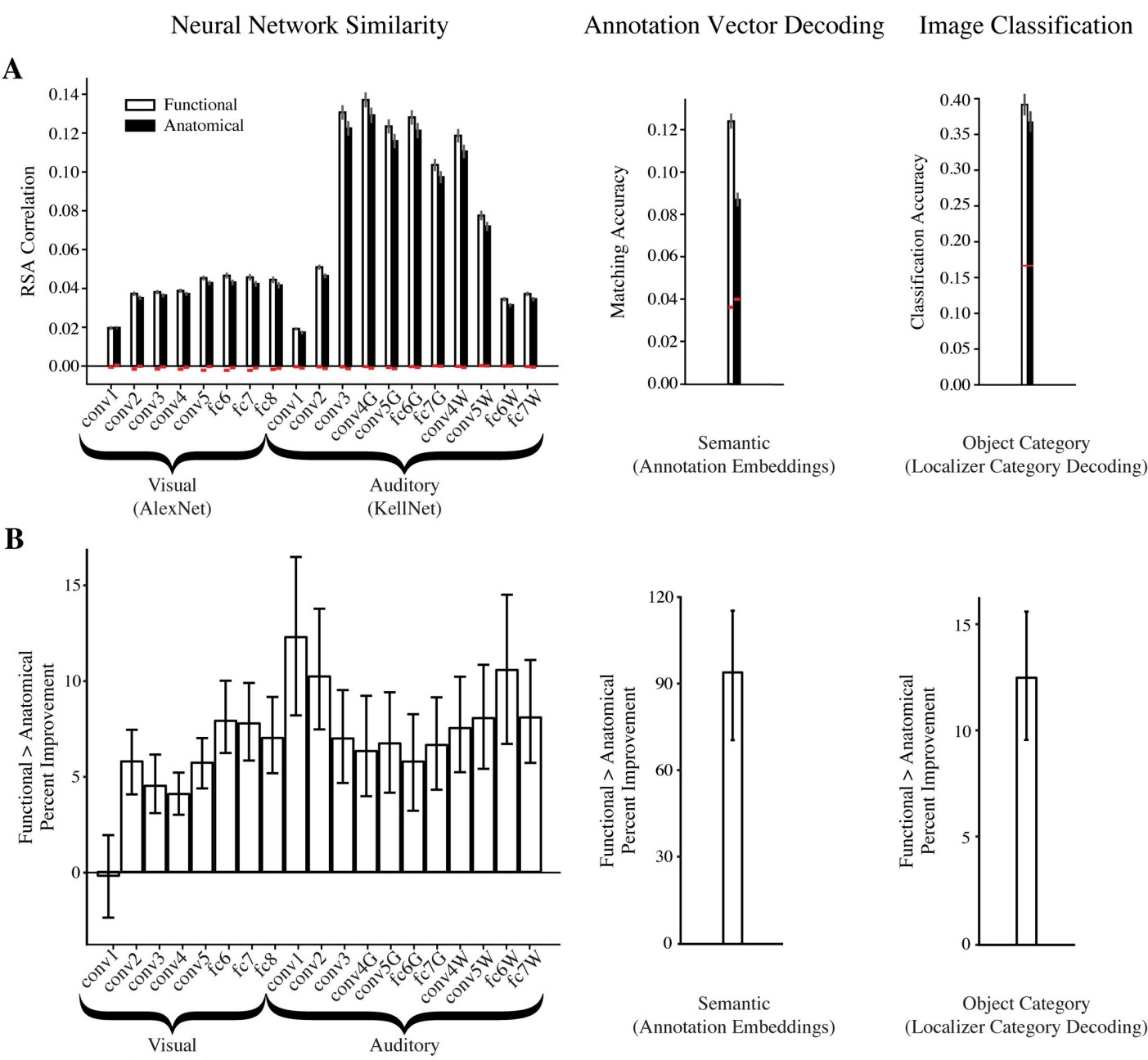

**Fig 3. Raw accuracy and percent improvement for all analyses.** (A) Average performance in the top 1% of functional and anatomical searchlights for neural network similarity, annotation vector decoding, and localizer category decoding. White bar is the functional searchlight performance, black bar is the anatomical searchlight performance. Error bars represent standard error across subjects. Chance (red lines) was computed in the neural network similarity and annotation vector decoding analysis as the mean of a null distribution estimated non-parametrically by rolling data in time, and in image classification as the theoretical chance level (1/ 6 categories). (B) Percent improvement of functional over anatomical searchlight in the top 1% of searchlights for neural network similarity (all layers), annotation vector decoding, and localizer category decoding. To calculate percent improvement, we first subtracted the chance level from the performance of each searchlight type. Error bars represent 95% CIs.

We performed follow-up analyses to explore the parameters that might affect functional searchlight performance and account for its consistent advantage over the anatomical searchlight. When we varied the searchlight radius (S4 Fig) the functional searchlight consistently

outperformed the anatomical searchlight, but the degree of this depended on the test performed, presumably due to the degree of distributed information required to succeed in the test. We also asked whether the advantage for functional searchlight is because bilateral information is being pooled, but even if the functional searchlight analysis was restricted within hemisphere, the functional searchlight outperformed the anatomical searchlight (S5 Fig). Finally, it is possible that the advantage of the functional searchlight is derived from having more voxels in the searchlight, since brain boundaries do not constrain the functional alignment method. However, when we yoke the two methods to have identical numbers of voxels in each searchlight, the functional searchlight is still superior (S6 Fig).

Although we used SRM to create the functional space in the main analyses, a lower-dimensionality representation can be learned in other ways, such as through principal components analysis (PCA). We explored replacing SRM with PCA in determining the functional searchlights. As with SRM, PCA functional searchlight outperformed anatomical searchlight (S7A Fig). However, SRM performed significantly better than PCA for the auditory and semantic analyses (S7B Fig), though not for the visual analyses. These results indicate that it is not always important to use a procedure that aggregates across participants (like SRM) in functional searchlight. In fact, because PCA is run within an individual, it can be used for any type of fMRI design (e.g., event-related trials in pseudo-random order across participants) and not just movie data where there is known correspondence in timing and content across participants. Nevertheless, SRM performed better than or equal to PCA almost across the board, suggesting that it is a good default when such correspondence exists.

To ensure that the utility of functional searchlight is not restricted to naturalistic movies, we analyzed the StudyForrest dataset [18], in which subjects watched the Forrest Gump movie and completed a block-design category localizer paradigm. We ran functional and anatomical searchlight analyses to decode localizer image category from voxel activity patterns (six categories: bodies, faces, houses, small objects, landscapes, scrambled images). As before, the functional space was learned from SRM of movie data (Forrest Gump). However, the resulting functional searchlights were now applied to a separate localizer task with blocks of the different categories. We found that the functional searchlight reliably outperformed anatomical searchlight (Fig 2C, S1 Table). The average increase in image classification was 12.576% (95% CI = [9.615, 15.917], bootstrap $p < 0.0001$). Even for this more standard form of image classification, which is often assumed to capture relatively localized representations in category-selective regions, the voxels that contributed to the top 1% of searchlights were much more distributed in functional searchlight than anatomical searchlight. (Fig 2D). Additionally, we tested whether the functional spaces generated from the Sherlock movie data and the Study-Forrest movie data were similar. We found that the pairwise distances between voxels in these spaces had a modest but highly significant positive correlation (S8 Fig). This suggests that SRM can capture a general organization of voxels common across audiovisual movie viewing conditions, but also that individual movie content may partly govern the learned functional spaces.

To confirm that running a searchlight in functional space can capture distributed information throughout the brain, we performed the RSA described above on simulated data with a known spatial signal distribution. In particular, we simulated fMRI data [19] where the signal from the units in the first fully connected layer of AlexNet was inserted into random voxels with varying degrees of spatial smoothness. Our results indicated that the relative improvement afforded by the functional searchlight is best when the signal is highly distributed throughout the brain (Fig 4). This supports our interpretation that the functional searchlight out-performs the anatomical searchlight because it picks up on information relevant for perceptual and cognitive processing that is distributed throughout the brain.

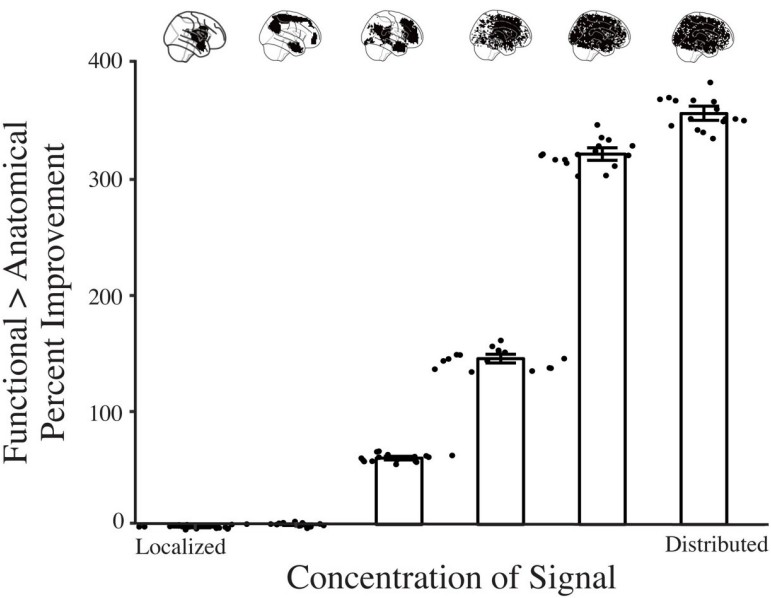

**Fig 4. Simulation of distributed versus localized representations.** We simulated fMRI data with varying degrees of localized signal. Signal localization was varied by sampling locations of signal voxels from a Gaussian Random Field (GRF) and varying its FWHM (x-axis). In particular, a GRF is simulated on the brain and the location of the signal voxels is determined by the highest values of the GRF. A GRF with high FWHM is smoother, so large values will tend to be clustered together. Therefore, for a GRF with high FWHM, the *n* highest values will be localized together. These highest values get more distributed as the FWHM decreases. We ran our RSA analysis for the first fully connected layer of AlexNet (see Methods). The error bars show 95% bootstrapped CIs and the dots represent individual subject improvements. As the signal transitioned from localized to distributed, the relative gain in performance of the functional searchlight increased.

## Discussion

Searchlight analysis, a common multivariate technique in fMRI based on anatomical volumes, is biased toward finding localized information. The functional searchlight is a way to better characterize distributed representations without assumptions about anatomical locality. The novelty of our method is both that it can be used to find a distributed set of voxels for this purpose (whereas standard anatomical searchlight cannot) and that functional searchlight outperforms anatomical searchlight in almost every case we considered. This approach revealed a tighter correspondence between the human brain and computational models as well as better results on a more standard image category decoding paradigm. An important methodological conclusion of this work is that conversion to functional space is worthwhile as a preprocessing step prior to searchlight analysis. Our results indicate that SRM is not absolutely necessary for creating the functional space because a within-subject dimensionality reduction method, such as PCA, can also be effective.

There are several other powerful methods for multivariate fMRI that can pick up on distributed representations. For example, regularized whole-brain methods [20] use all voxels in the brain as features for a regression but put a penalty on the matrix norm of the regression's feature weights. In other words, all brain voxels are considered, but the space of solutions of how voxels are weighted is restricted to those in which the vast majority of weights are either very small or 0 and only a sparse set of voxels are the main drivers of the regression. Sparse-overlapping Sets (SOS) Lasso extends this idea using structured sparsity [21]. In SOS Lasso, neighboring voxels within a specific radius are grouped into sets and, instead of having weights on individual voxels that contribute to the sparsity penalty, weights are put on these voxel sets for

the sparsity penalty. PrAGMATiC [22] is a similar approach that instead uses probabilistic groupings.

These powerful methods generally utilize free parameters that need to be selected through cross-validation or other computationally intensive routines. For example, regularized whole-brain methods [20] require choosing a regularization function (L1 or L2) and fitting of a penalty parameter, both of which have to be optimized through cross-validation grid search. SOS Lasso also has choices for penalty terms and free parameters [21]. PrAGMATiC [22] requires the fitting of extra parameters through a Markov Chain Monte Carlo (MCMC) framework. Functional searchlight with SRM does not involve any free parameters, nor assumptions about the sparsity of representations in the brain. That said, it does assume that the stimulus used to construct the functional space will rearrange the voxels in a way that is relevant for subsequent analyses. It is possible that in some domains, such as for complex cognitive control tasks, rearranging voxels based on passive movie viewing will not improve analyses of those tasks. Future work should investigate the boundary conditions of tasks that can benefit from functional searchlight.

By enhancing model-based analysis in this way, fMRI can inform the development of new, biologically plausible computational frameworks of cognitive function. This work also has theoretical implications in suggesting that, during viewing of a naturalistic audiovisual stimuli, the brain's representations of visual, auditory, and semantic information encoded within deep neural network and natural language processing models, as well as object representations, are more widely distributed than previously thought.

## Methods

### Ethics statement

This work is a computational experiment on public, anonymous data. Two public datasets were used in this work [11,18]. The studies that generated these datasets were approval by the local institutional review boards, as detailed in the original publications. No additional ethics approvals were needed or obtained for the re-analyses reported herein.

### General approach

**Searchlight analysis.** We conducted functional and anatomical searchlight analyses on two different public datasets: 'Sherlock' [11] and 'StudyForrest' [18] and evaluated the difference in top-performing voxels across the two methods. The analyses on the 'Sherlock' dataset were based on visual (AlexNet RSA), auditory (KellNet RSA), and semantic (annotation embedding decoding) content of the movie while the analysis on the 'StudyForrest' was on standard category decoding of still localizer image stimuli.

Searchlight analysis was performed by iterating the same computation over subset volumes of voxels across the brain. The BrainIAK package was utilized for parallelizing the computations [23]. Each searchlight was a tensor centered on every voxel inside the brain, with a radius of 3 voxels. This tensor was thus 7 x 7 x 7 x $t$ such that it represented a cube of voxels and $t$ time points from the movie (for 'Sherlock': $t$ = 946 or 1030 for part 1 and part 2, respectively; for 'StudyForrest': t = 624). When near the edge of the brain, anatomical searchlights often contained fewer than the full 343 voxels of the cube because non-brain voxels were excluded (minimum for 'Sherlock' = 115, 'StudyForrest' = 97). In the resulting brain map, each voxel's value represents the output of the analysis on that searchlight's voxels. In this study, we refer to this traditional searchlight analysis as an anatomical searchlight. We use the anatomical searchlight as a baseline to compare with our new functional searchlight approach.

In the functional searchlight, data were first embedded into a space where distances between voxels were based on functional distance rather than anatomical distance, and then a searchlight was run on these transformed data. Voxels that represent similar content will be close to one another in functional space and thus included in the same searchlight. To create this functional embedding, we use SRM [8], which maps voxels into a lower-dimensional feature space of fMRI activity that reflects what is shared across participants while they view a common stimulus. SRM is an unsupervised dimensionality reduction technique used for functionally aligning multiple subjects' fMRI data together on a dimensionality-reduced shared space. We model fMRI responses $X$ ($v$ voxels by $t$ times data matrix) for subject $i$ as $X_i = W_i S + E_i$. $W_i$ ($v$ by $k$ features weight matrix) is subject-specific orthonormal bases representing individual functional topographies. $S$ ($k$ by $t$ shared response matrix) is latent features that capture shared variance across all subjects. $E_i$ ($v$ by $t$ error matrix) is the subject response not captured by the shared response to the common stimulus. SRM learns $n$ (subjects) $W_i$ orthogonal weight matrices and one $S$ shared response matrix such that the quantity $\sum_{i=1}^{n} ||X_i - W_i S||_F$ is minimized.

SRM can be applied to data in which participants are engaged by a common stimulus. In our case, all subjects viewed the same audiovisual stimulus ('Sherlock': 1973 TRs, 'StudyForrest': 3599 TRs). For each dataset, SRM is trained on a separate portion of the data from what was used for the main analyses. For 'Sherlock', we divided the data into two parts, the first with 946 TRs (part 1) and the second with 1030 TRs (part 2). We trained the SRM on part 1 and ran our analyses on part 2, then reversed this order to train the SRM on part 2 and run analyses on part 1. We averaged the results of both analysis folds. For 'StudyForrest', the movie viewing portion (3599 TRs) was already separate from the localizer experiment portion (624 TRs) on which we performed our searchlight analyses. For training the SRM, we only used the first two runs of the movie portion (892 TRs) to roughly match the number of TRs we used to train the SRM on the Sherlock dataset.

SRM is used to create a $k$-dimensional shared feature space, where $k$ is selected via time-segment matching (see below). This means that each voxel had a weight, from their subject-specific orthogonal weight matrix, for how much that voxel loads on to each of the $k$ features. These weights were then used as coordinates to remap that voxel into a $k$-dimensional space, independent of the voxel's anatomical location. Voxels with a weight of 0 for each of the $k$ features were discarded, due to them not contributing at all to the shared feature space. We then defined a searchlight for each voxel in functional space as the $j$-nearest neighbors of that voxel, where distance was defined by the cosine distance between voxels' functional space coordinates. We chose $j$ to match the size of the anatomical searchlights (in this case, $j = 342$). With the exception of voxels that have exactly zero weight to the shared space, *all* voxels are used in the functional searchlight, not just voxels with high shared space weights.

**Statistics.**   For our primary analysis (Fig 2), we obtained the average performance of the top 1% of searchlights for both functional searchlight and anatomical searchlight approaches. We then subtracted the empirically derived chance performance for each searchlight type (see below) from these averages and calculated the percent improvement of functional searchlight over anatomical searchlight. To test significance, we generated 95% CIs of percent improvement by bootstrap resampling across participants [24]. Specifically, we sampled with replacement from the percent improvements of each of the ($n = 16$ for 'Sherlock', $n = 15$ for 'StudyForrest') subjects, calculated the average of each sample, and repeated the process 10,000 times. The logic of this approach is that to the extent that the effect is reliable across participants, the participants should be interchangeable and it should not matter which subjects are sampled on a given iteration. We report the 95% CIs as the 2.5th percentile and 97.5th percentile of the resampled distribution. Functional searchlight performed significantly better than

anatomical searchlight if and only if the lower bound of the CI for percent improvement is above 0.

For 'StudyForrest', since the analysis had an *a priori* chance performance level (image category decoding), chance was set as 1/number of categories (in this case, 1/6). For 'Sherlock', since these analyses did not have *a priori* chance performance levels (e.g. neural network RSA), we empirically determined the chance level non-parametrically using a rolling analysis. Across 100 iterations, we misaligned the brain and model data in time by shifting one of them with respect to the other in multiples of 10 timepoints. By repeating all analyses after such misalignment, we populated a null distribution of performance. We defined chance performance for a given subject and analysis as the mean of this null distribution. This was then used as a reference for the real value, to calculate performance above chance.

## Sherlock dataset

**Participants.** Full details can be found in the original publication of this dataset [11]. We included the 16 participants who viewed the entire movie. They were right-handed, had normal or corrected-to-normal vision, and had never seen BBC's Sherlock previously. Informed written consent was obtained in accordance with a protocol approved by the Princeton University IRB.

**Materials.** Participants watched a 48-minute clip of the BBC television series "Sherlock" from Season 1, Episode 1. The stimulus was divided into two similarly sized segments (part 1: 946 TRs and part 2: 1030 TRs). At the beginning of each segment, an audio-visual cartoon was displayed for 30 s.

**Data acquisition and preprocessing.** The fMRI data were collected on a 3-T scanner (Siemens Skyra) with a 20-channel head coil. A $T2^*$-weighted echo-planar imaging (EPI) pulse sequence (TE 28 ms, TR 1500 ms, flip angle 64˚, 27 slices, 4 mm thickness, $3 \times 3$ mm in-plane resolution, FOV $192 \times 192$ mm, whole-brain coverage) was used to acquire functional images. A T1-weighted MPRAGE pulse sequence (0.89 mm isotropic resolution) was used to acquire anatomical images. Preprocessing included slice-time correction, motion correction, linear detrending, high-pass filtering (140 s cutoff), and registration of the functional volumes to a template brain (MNI) at 3mm resolution. Every voxel was z-scored in time within movie run and timing was aligned across participants.

**Time-segment matching.** We ran a time-segment matching analysis [8] to select the number of dimensions for the functional space. An SRM was fit on one half of the training data (i.e., one quarter of the episode) and used on the second half of the training data in a leave-one-participant-out cross-validation. Time segments of 14s were taken from the left-out participant's data and correlated with all segments of the same length in the shared space averaged over the remaining participants. The time-segment matching accuracy for each subject was the proportion of test time segments that were maximally correlated with the correct time segment in the shared space. We varied the number of features from 3 to 400 (step size of 10) and observed that accuracy plateaued in both parts around 200 features (S1 Fig). Given these results, we used $k = 200$ dimensions for the functional space.

**DNN similarity analysis.** We compared the similarity of activity patterns within functional and anatomical searchlights to the similarity of visual and auditory representations derived from deep neural network models. In particular, we compared a time-point by time-point RSA of searchlight data to the similarity structure of visual (AlexNet [11]) and auditory (KellNet [12]) DNNs also viewing the movie. AlexNet is a visual object recognition network with five convolutional layers (conv1-5) and three fully connected layers (fc6-8). It takes a 227x227x3 colored image and outputs a 1,000-unit vector in its last fully connected layer,

which contains its confidence of the image belonging to one of 1,000 image categories. KellNet is an auditory network with separate branches for word recognition and music genre recognition. It takes as input a cochleagram generated from an audio waveform and returns a word category and music genre category in each respective branch. The two branches share three convolutional layers (conv1-3) and then each have their own two convolutional layers (conv4/5_W for word recognition and conv4/5_G for music genre recognition) as well as their own two fully connected layers (fc1/top_W for word recognition and fc1/top_G for music genre recognition). fctop_W is a 587-unit vector that contains confidence in the sound belonging to one of 587 word categories. fctop_G is a 41-unit vector that contains confidence in the sound belonging to one of 41 music genres. In order to keep the naming convention consistent with AlexNet, we refer to fc1_W and fc1_G as fc6W and fc6G, respectively, along with fctop_W and fctop_G as fc7W and fc7G, respectively.

To obtain the model-based activity for the visual modality, AlexNet received individual frames of the movie. We extracted the activations from each convolutional layer and fully connected layer. We averaged the activity across a subset of frames within the same TR so that the output was a pattern of activity for each TR of the movie. Each TR lasted 1.5 seconds and the movie had 25 frames per second, so a single TR theoretically contained 37.5 frames. We averaged activity using the middle 25 frames of each TR to reduce the autocorrelation across DNN features of different TRs. The end result was AlexNet activity of the entire Sherlock movie in TR intervals. To obtain the model-based activity for the auditory modality, we broke up the movie into 1.5s audio clips (to match the TR) and generated cochleagrams for each TR to be used as an input to KellNet. The activity these clips generated at each of the convolutional layers was then stored. This model-based activity was used to create a representational similarity matrix (RSM) for each layer. Let $t$ be the number of TRs in the movie. For each layer and for each model, we constructed a $t$ x $t$ correlation matrix by correlating unit activity vectors across different TRs.

For each subject, we also constructed an RSM within each searchlight of the brain by correlating voxel activity across different TRs. In these brain RSMs, elements close to the diagonal will be highly similar due to the autocorrelation of the BOLD response. To remove this potential confound in measuring representational similarity, we imposed a buffer of 10 TRs (15s), such that we only retained correlations between TRs separated by at least this much time. These retained off-diagonal elements (i.e., TR pairs) of the model RSM and brain RSM were unraveled into vectors of the same length. We then correlated these vectors to quantify the similarity between representations in each searchlight and in a given layer of a computational model, and assigned this second-order correlation to the center voxel.

**Natural language decoding analysis.** We compared functional and anatomical searchlight performance on a natural language decoding analysis. The Sherlock dataset contains a handwritten annotation of what is happening in each TR of the movie (one sentence per TR, averaging 18 words). In previous work, natural language processing (NLP) techniques were used to create semantic vector representations of each annotation [15]. That is, each TR was represented by a 100-dimensional vector that encodes the semantic meaning of the corresponding annotation. These vectors were constructed from word embeddings built using a latent-variable modeling approach [15,25] on the Wikipedia corpus for each word in a given annotation. A domain-specific re-centering was applied to these word embeddings to make each of them discriminative within the average topic of the Sherlock annotation vocabulary. An embedding was defined for the entire annotation as a weighted average of the constituent word embeddings, with weights determined by relative word frequencies.

We predicted these annotation vectors from brain activity using a scene ranking analysis [15]. We used the first half of the movie to train the SRM and create the functional space, which was then applied to the second half of the movie prior to analysis. Within the second

half, we trained a ridge regression model on half of the timepoints (i.e., quarter of the movie) to take searchlight voxels and predict the 100-dimensional annotation embedding of the other half of timepoints. Specifically, we trained the SRM and learned the functional space on part 1, then trained the ridge regression model on the first 530 TRs of part 2 to predict the last 500 TRs of part 2. This resulted in a predicted 100-dimensional annotation vector for each of these 500 TRs, which could be compared against the actual annotation vectors for these TRs to quantify performance. We divided the 500 predicted and actual annotation vectors into 25 evenly sized bins, comprising 20 annotation vectors from consecutive TRs. We then computed a 25 x 25 correlation matrix $M$, where $M_{ij}$ denotes the correlation of predicted bin $i$ with the actual bin $j$. Our reported accuracy is the proportion of predicted bins that were most highly correlated with the corresponding actual bin (chance accuracy = 1/25 or 4%). Unlike previous analyses, we did not reverse the order and train SRM on the second part and do analysis on the first part. This is because the annotation vectors we used from [15,25] were normalized using data from part 1 of the movie, so to avoid double dipping, we refrained from doing annotation vector decoding on part 1 of the movie.

### Study forrest dataset

**Participants.** Full details can be found in the original publication of this dataset [18] and on the data sharing website: http://studyforrest.org/. We included the 15 participants who completed both the movie viewing and localizer tasks. They were right-handed, had normal or corrected-to-normal vision, and were native German speakers. All but three participants had previously seen the movie *Forrest Gump*. They signed an informed consent for public sharing of data in anonymized form. This study was approved by the Ethics Committee of the Otto-von-Guericke University.

**Materials.** Participants watched the 2h (3599 TRs) movie *Forrest Gump* while listening to the German audio track of the movie. In the specific StudyForrest dataset extension we used [18], participants also completed a functional localizer task (624 TRs). They viewed 24 unique grayscale images from six categories (human faces, human bodies without heads, houses, small objects, outdoor scenes, and phase-scrambled images) in a block design. There were four localizer runs, each of which contained two blocks per stimulus category. Each block contained 16 images presented one-at-a-time for 900 ms followed by 100 ms of intertrial interval (ITI). To maintain attention, participants performed a one-back task on the images.

**Data acquisition and preprocessing.** Extensive details on fMRI acquisition and preprocessing are available in the original "StudyForrest" publication [26] on the functional movie data and anatomical scans and the extension publication [18] on the functional localizer data. The movie data were collected on a whole-body 7-Tesla Siemens MAGNETOM MRI scanner equipped with a local circularly polarized head transmit and a 32-channel brain receive coil. A T2*-weighted echo-planar imaging (EPI) pulse sequence (gradient-echo, TR 2s, 90˚ flip angle, 22 ms echo time, 0.78 ms echo spacing, 36 axial slices, 1.4 mm thickness, 1.4 x 1.4 mm in-plane resolution, FOV 224 x 224 mm, whole-brain coverage) was used. EPI images were corrected online for geometric and motion distortion. The localizer data were collected on whole-body 3-Tesla Philips Achieva dStream MRI scanner equipped with a 32-channel head coil [18]. A T2*-weighted EPI pulse sequence (gradient-echo, TR 2s, 90˚ flip angle, 30 ms echo time, 35 axial slices, 3.0 mm thickness, 3.0 x 3.0 mm in-plane resolution, FOV 240 x 240 mm, whole-brain coverage) was used. Preprocessing in the original work included a bandpass filter (retaining 9–150 s period) and z-scoring within voxel.

Structural scans were obtained with a 3-Tesla Philips Achieva MRI scanner equipped with a 32-channel head coil (same as functional localizer task) using standard clinical protocols. A

T1-weighted image was acquired with a 3D turbo field echo (TFE) sequence (TR 2500 ms, TE 5.7ms, 8˚ flip angle, 274 sagittal slices, 0.67 isotropic resolution, FOV 191.8 x 256 x 256 mm, whole-brain coverage). We used the structural scans to align each participant to standard MNI space using FLIRT in FSL [27].

**Localizer category decoding analysis.** We used both components of the StudyForrest dataset. The first two runs of the movie data (892 TRs) were used to train an SRM model for defining the functional space. To match the Sherlock dataset, we set $k = 200$ features for SRM. The localizer data were then used for MVPA of image category within functional and anatomical searchlights. In both analyses, the searchlight kernel function used was a 3-fold stratified cross-validation analysis, in which we trained a linear 6-way SVM classifier in each fold to decode category. Performance for each searchlight is reported through mean accuracy across folds. Because the training and test data of each fold are stratified, each image category is equally represented and so the chance accuracy of this analysis is $1/6 = 16.67\%$ (1/number of categories).

### Simulated dataset

We assume that functional searchlight outperforms anatomical searchlight because it can capture distributed information. We evaluated this interpretation by simulating fMRI data that varied in how broadly information was distributed across the brain and repeating the AlexNet analysis in the 'Sherlock' experiment. The simulation was performed using the fmrisim package in BrainIAK [19]. In particular, we simulated realistic fMRI noise from participant templates [28] and chose a set of voxels in the simulated brain to insert signal. We simulated 16 participants whole-brain data and inserted signal from the 4096 units of fc6 in AlexNet into 4096 voxels. Hence, if there were voxels in the brain that responded exactly the same the fc6 layer of AlexNet then their activity would look something like the signal that was simulated. This signal was generated by convolving the unit activity (averaged across frames) for each TR with a double-gamma hemodynamic response function. This signal was then added to the realistic neural noise so that the overall timeseries contained signal from the deep network, while retaining properties of BOLD such as autocorrelation and lag. The magnitude of the inserted signal was set at 0.5 percent signal change.

The voxels chosen to carry signal were determined based on how distributed the induced signal was. In particular, we simulated a Gaussian Random Field (GRF) the same shape as each simulated participant's brain and chose the voxels containing the highest 4,096 GRF values as the locations of the signal voxels. The GRF has a specific smoothness, parameterized by the full-width half max (FWHM). If the GRF is smooth, then all of the highest values are likely to be localized in a single area. However, if the GRF is not smooth, then these signal voxels will be distributed throughout the brain. We created volumes with signals distributed over FWHMs of 0.5, 1.0, 2.0, 4.0, 8.0, 16.0. In other words, we simulated participant data with signal distributed locally or broadly throughout the brain. To quantify performance, we repeated the main representational similarity analysis of AlexNet on simulated data from anatomical and functional searchlights.

### Supporting information

**S1 Fig. Selecting dimensionality of SRM with time-segment matching.** We performed a time-segment matching analysis [8] on the Sherlock data to determine how many SRM dimensions to use. Within the training half of the movie, we trained an SRM with the corresponding number of dimensions ($x$ axis) on one half of the TRs and tested on the second half of the TRs. The goal was to predict from which time window in the movie the test data were obtained

(chance = 0.0212). Time-segment matching proportion correct (*y* axis) plateaued at approximately 200 dimensions considering both folds of the overall analysis.
(TIF)

**S2 Fig. Distributed representations in individual subjects.** The spatial distribution of the top 1% functional searchlight voxels for each subject and analysis.
(TIF)

**S3 Fig. Anatomical distance between voxels within searchlights.** The median Euclidean distance (in voxel units) of the anatomical coordinates of voxels from within the top 1% of functional searchlights in the movie content experiment. Error bars denote 95% CIs across subjects and the red line denotes the median Euclidean distance of voxels from within anatomical searchlights (constant determined by searchlight radius). An appropriate ceiling for these values is the median Euclidean distance in a searchlight consisting of randomly chosen brain voxels (approximately 28 voxels).
(TIF)

**S4 Fig. Effect of searchlight size on advantage of functional searchlight.** For our main analyses, we chose a searchlight radius = 3 (number of voxels for each searchlight 7 x 7 x 7 = 343). Here we report movie content analyses results for radius = 2 (top row; number of voxels: 5 x 5 x 5 = 125) and radius = 4 (bottom row; number of voxels: 9 x 9 x 9 = 729). A larger radius led to a greater advantage for functional over anatomical searchlights in neural network analyses and a smaller advantage for the NLP analyses. The y-tick values in the top plot for the semantic analysis are a different range because the anatomical searchlight on the natural language embeddings does very poorly when the searchlight radius is low. This speaks to our finding that these natural language representations are widely distributed rather than localized in circumscribed regions of the brain.
(TIF)

**S5 Fig. Within-hemisphere functional searchlight analysis.** To test whether the improved performance of functional searchlights reflects their ability to aggregate information across hemispheres from bilateral regions that are anatomically distant but functionally homologous (e.g., left and right auditory cortex), we re-ran all of our movie content analyses within one hemisphere of the brain at a time (including fitting SRM) by completely ignoring the other hemisphere. The error bars show 95% bootstrapped CIs. Even in the absence of bilateral information, functional searchlight still consistently outperforms anatomical searchlight.
(TIF)

**S6 Fig. Equating the number of voxels per searchlight across searchlight types.** Some anatomical searchlight volumes on the edge of the brain would contain non-brain voxels that need to be excluded, resulting in fewer brain voxels defining the activity pattern. Since we are taking the nearest neighbors in functional space, the number of voxels per functional searchlight is always the same. To ensure functional searchlight is not benefitting from having more voxels in certain searchlights, we ran a version of our movie content analyses where each functional searchlight was forced to contain the exact same number of voxels as the anatomical searchlight for that voxel. The error bars show 95% bootstrapped CIs. Even when we force the number of voxels per searchlight to match, functional searchlight still performs better.
(TIF)

**S7 Fig. Substituting PCA when defining functional space.** We repeated our analyses on the movie content experiment but replaced SRM with PCA. In particular, we trained a PCA within each subject using the half of the movie that was used for SRM training. Equating the number

of dimensions used for SRM, we took the top 200 components (ordered by explained variance) and used each voxel's loading onto the components as coordinates for functional space. We compared PCA functional searchlight vs. anatomical searchlight (A) as well as SRM functional searchlight vs. PCA functional searchlight (B). PCA functional searchlight outperformed anatomical searchlight. Defining functional space with SRM and PCA yielded similar results for the visual network similarity analyses, but SRM outperformed PCA on both auditory network similarity and the semantic analyses.
(TIF)

**S8 Fig. Comparing SRM functional spaces from different datasets.** To determine the similarity of the functional spaces constructed by SRM for the Sherlock and StudyForrest datasets, we extracted the Euclidean distances between all voxel pairs in functional space and then correlated these distances between datasets. The overall correlation was estimated by randomly sampling the same 10,000 voxel pairs from the two datasets, calculating the correlation of distances in the sample, repeating 1,000 times with new random samples, and then averaging across iterations. There was a positive correlation overall ($r = 0.198$). To test statistical significance, we performed a non-parametric randomization test by repeating this process 1,000 times while randomly permuting the voxel distances of the Sherlock dataset each time to populate a null distribution. The true correlation of the datasets was highly significant ($p < 0.001$). One of the samples used to estimate the overall correlation is visualized in the plot, with each dot depicting one voxel pair and the axes corresponding to their Euclidean distance in the two functional spaces. Color coding by anatomical distance of each voxel pair indicates how dramatically SRM reshaped the brain based on function (i.e., voxels that are anatomically distant can be functionally close, and vice versa). a.u. indicates arbitrary units.
(TIF)

**S1 Table. Quantification of functional searchlight performance across all analyses.** These statistics were calculated based on the percent difference of the mean performance of the top 1% functional searchlights over the mean performance of the top 1% anatomical searchlights, using empirically derived chance performance as a baseline.
(DOCX)

## Acknowledgments

We thank Janice Chen and her colleagues in the laboratories of Ken Norman and Uri Hasson at Princeton University for collecting and sharing the Sherlock dataset, as well as the Psychoinformatics lab at Otto-von-Guericke University Magdeburg for collecting and sharing the StudyForrest dataset. We further thank Ken Norman for helping us obtain the annotation embeddings.

## Author Contributions

**Conceptualization:** Sreejan Kumar, Cameron T. Ellis, Nicholas B. Turk-Browne.

**Data curation:** Sreejan Kumar.

**Formal analysis:** Sreejan Kumar.

**Funding acquisition:** Marvin M. Chun, Nicholas B. Turk-Browne.

**Investigation:** Sreejan Kumar, Cameron T. Ellis, Thomas P. O'Connell.

**Methodology:** Sreejan Kumar, Cameron T. Ellis, Thomas P. O'Connell, Nicholas B. Turk-Browne.

**Project administration:** Nicholas B. Turk-Browne.

**Resources:** Marvin M. Chun, Nicholas B. Turk-Browne.

**Software:** Sreejan Kumar.

**Supervision:** Marvin M. Chun, Nicholas B. Turk-Browne.

**Validation:** Sreejan Kumar, Cameron T. Ellis.

**Visualization:** Sreejan Kumar.

**Writing – original draft:** Sreejan Kumar, Cameron T. Ellis, Nicholas B. Turk-Browne.

**Writing – review & editing:** Sreejan Kumar, Cameron T. Ellis, Thomas P. O'Connell, Marvin M. Chun, Nicholas B. Turk-Browne.

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
