## [Decision Letter · Decision Letter 0]

2 Jul 2020

Hi - As you will see, one of the reviewers has very strong reservations regarding the significance of the method. Please try to address these and other points raised by both reviewers in your rebuttal.

Best wishes

Saad.

Dear Dr. Turk-Browne,

Thank you very much for submitting your manuscript "Searching through functional space reveals distributed visual, auditory, and semantic coding in the human brain" for consideration at PLOS Computational Biology.

As with all papers reviewed by the journal, your manuscript was reviewed by members of the editorial board and by several independent reviewers. In light of the reviews (below this email), we would like to invite the resubmission of a significantly-revised version that takes into account the reviewers' comments.

We cannot make any decision about publication until we have seen the revised manuscript and your response to the reviewers' comments. Your revised manuscript is also likely to be sent to reviewers for further evaluation.

Sincerely,

Saad Jbabdi

Associate Editor

PLOS Computational Biology

Daniele Marinazzo

Deputy Editor

PLOS Computational Biology

Reviewer's Responses to Questions

**Comments to the Authors:**

Reviewer #1: Authors in this paper propose an alternative to traditional searchlight based search for information called functional searchlight. Gist of it is to use a set of voxels that have high weight in some latent shared dimensions that explain most variance during a movie viewing task. They show that these sets of voxels as opposed to anatomically selected sets have higher correlation with representations of different layers of DNNs for audio and visual input and higher accuracy in decoding movie scenes.

Manuscript is clearly written and well structured and easy to follow.

I have major concerns about the proposed method, its usability in many scenarios compared to traditional searchlight, and the claim of distributed representation being confounded by the stimulus used.

Proposed method:

- Authors present this as a novel idea/method but this has been presented and used by the original papers on Hyperalignment: Haxby et al (2011) restricted to VT and then expanded to whole brain in Guntupalli et al (2016)

- Using top-weighting voxels in the reduced shared space and treating them as unit of information representation is problematic for two reasons: 1. these could be driven by correlations in the movie, which are by design correlated across domains, 2. without showing that these are truly distributed representations, one needs to show the evidence in an experiment devoid of such correlated stimulus. For example, in the above mentioned papers, they do so on controlled independent experiments with still images, functional localizers, and retinotopy. Extraordinary claims like this require extraordinary evidence.

- As authors acknowledge in their introduction, people have done whole brain decoding before and as they rightly pointed out, due to curse of dimensionality most methods rely on dimensionality reduction methods such as PCA/ICA etc. Here, they used SRM, a variant of hyperalignment, to reduce dimensionality, but do not explain why they do that instead of, say, using PCA of every individual's data and showing that high-weighting voxels for top-N PCs are distributed (which they will be). I understand the reasons due to my expertise but it is not clear in the paper.

Utility of the method:

- Searchlight as originally proposed is useful and popular for good reasons. It is comparable across studies and not conflated by correlations in the stimulus.

- I am not sure how the method proposed here can be used by a generic fMRI study. Authors do not give any evidence that these functional SLs are meaningful and comparable across studies.

- If the idea is to use them for decoding or stimulus feature modeling, then it has already been proposed and used elsewhere and there is nothing novel about that.

- Anatomical/functional ROIs provide natural regularization and therefore generalize beyond the training distribution. Authors provide no evidence that their method generalized beyond training distribution. No, part 2 of same movie doesn't count. They both have similar correlations.

Problems with claims:

- SRM and the original hyperalignment are known to clean up signal related to shared stimulus content, acting like a filter. To give a hypothetical example, imagine using a stimulus with no audio for deriving shared model, it invariably reduces weights to voxels in auditory regions of STS, this will improve SNR for subsequent analyses. So, it is important to compare anatomical against SRM filtered data: Guntupalli et al. (2016) did that by retrojecting the hyperaligned data into back into subjects (see their papers for specifics).

- Another problem is with decoding movie scenes. If we only look at a SL in calcarine sulcus, we will not be able to differentiate two scenes with same visual but if we include voxels in Heschl's gyrus, we can use any discriminatory auditory information to differentiate the two. So, the results about better decoding of movie scenes with voxels distributed across the brain compared to the same number of localized voxels is trivial and shown before in aforementioned papers with the acknowledgement of the reasoning.

Overall, there are many issues which the authors fail to acknowledge or present in the paper. Discussion section section should cover these in detail so the readers know the caveats.

Minor comments:

- Results are not reported in text beyond qualifiers. Please provide the numbers and statistical test results in test and actual measures to assess the effect sizes.

Figure 2: Mentioning actual correlation values for RSA and decoding in caption will give indication effect sizes.

Figure S4 top right panel’s Y-tick values seem wrong

- All the analyses the authors did were done in prior work, please acknowledge that in introduction.

Reviewer #2: This paper describes a new method for finding distributed representational structure in functional brain imaging data, and compares it to the widely-used anatomical searchlight approach. The new method first maps BOLD data into a low-dimensional functional space using shared-response mapping, so that voxels showing similar functional profiles are situated near one another in the functional space. They then apply the searchlight technique, using the functional proximities to define searchlights rather than anatomical proximities. The authors use anatomical and functional searchlights to decode visual, auditory, and semantic structure in the Sherlock dataset, comprising functional brain images from 16 subjects as they watched an episode of the TV show Sherlock. They show that the new method can better predict target similarity matrices in all cases, and suggest a more anatomically distributed signal relative to the standard approach, consistent with a developing view that neuro-cognitive representations are more distributed in the brain than classical view have suggested.

This work contributes to a developing literature innovating new data-science approaches to learn more about how information is encoded in neural systems. As the authors note in the introduction, a long-standing debate in cognitive neuroscience concerns the degree to which different functions should be viewed as anatomically localized or distributed. In brain imaging, textbook findings consistent with localizationist views may be an artifact of widely adopted statistical methods that are only capable of finding localized signal, a point that has been emphasized by a variety of groups in recent years. The approach to finding less localized signal in the current work was new to me, however, and represents a potentially important contribution to this literature. The paper is very clearly and concisely written, and the results seem compelling. I have some suggestions for further clarifying and elaborating the approach and the findings that I hope may be useful in maximizing the paper's impact.

1) The results are reported in terms of percent improvement over the anatomical searchlight method, but it would also be useful to see the empirical model accuracy. Often searchlight methods report results that are statistically distinguishable from a null result, but nevertheless weak (e.g. correlation between neural and target RDM of ~0.05, reliably non-zero but....). It would be useful to see what the actual correlation between predicted and true RDMs are for the top 1% of searchlights here. I understand that the point of the paper is to show that functional searchlight is reliably better-performing than anatomical, but for others adjudicating whether to adopt this or other approaches, the raw fits would be useful to see.

2) A central finding is the more broadly-distributed signal revealed by the functional-searchlight result, which also shows greater accuracy. This is especially noteworthy as both the size of the searchlight and the number of "good performing" searchlights are matched between functional and anatomical approaches---so the broader signal does not reflect, for instance, more searchlights being "selected." But, I think there are two importantly different potential explanations. One is that searchlights in all subjects are grouping together anatomically-distal voxels; the other is that the selected voxels vary quite alot in their location across subjects. It would be useful to know how consistent the anatomical distribution of selected voxels is across subjects.

3) While the paper is mainly very clearly written, I didn't understand what information was contained in the "S" matrix. The authors state (p. 13) that S is a k-by-t matrix "representing the shared response". It is clear that the SRM is encoding each individual subject's time-series data as a weighted combination of a common set of features at each time-point, but I don't know what the "shared-response" features are or how they were computed. Also, I think there is a typo in this paragraph, which describes W as a k-by-v matrix and S as a k-by-t matrix. I think one of those needs to be inverted in order to multiply these (like, maybe V is v-by-k).

4) It would be useful to see some discussion of how this approach relates to other approaches that likewise aim to find widely-distributed representational structure, such as plain old whole-brain regularized regression (as explored, for instance, by Irina Rish), the "generative modeling" approach pursued by Tom Mitchell, Jack Gallant etc, and the "structured sparsity" approaches (SOS LASSO, network RSA) developed by our group. I realize extensive discussion is out of bounds for a short paper, but a paragraph briefly sketching how the current approach relates to these would be v helpful.

**Have all data underlying the figures and results presented in the manuscript been provided?**

Reviewer #1: Yes

Reviewer #2: Yes

PLOS authors have the option to publish the peer review history of their article (what does this mean?). If published, this will include your full peer review and any attached files.

Reviewer #1: No

Reviewer #2: **Yes: **Timothy Thomas Rogers
---

## [Decision Letter · Decision Letter 1]

14 Oct 2020

Dear Dr. Turk-Browne,

See below minor additional comments from one reviewer.

Best wishes

-----

Thank you very much for submitting your manuscript "Searching through functional space reveals distributed visual, auditory, and semantic coding in the human brain" for consideration at PLOS Computational Biology. As with all papers reviewed by the journal, your manuscript was reviewed by members of the editorial board and by several independent reviewers. The reviewers appreciated the attention to an important topic. Based on the reviews, we are likely to accept this manuscript for publication, providing that you modify the manuscript according to the review recommendations.

Sincerely,

Saad Jbabdi

Associate Editor

PLOS Computational Biology

Daniele Marinazzo

Deputy Editor

PLOS Computational Biology

[LINK]

Reviewer's Responses to Questions

**Comments to the Authors:**

Reviewer #1: Thanks for addressing my concerns and taking pains to run control experiments and clarify some misconceptions such as signal filtering.. I apologize if my comments came out strong which wasn't my intention but to express my concerns with the interpretation and claims of novelty of some aspects of the paper.

Few minor questions: how do the functional searchlights look across the two studies (Forrest and Sherlock). Do they reveal similar distributed patterns across studies? This is related to my original concern of whether these are revealing some general distributed networks comparable across experiments.

- Results section still doesn't contain numbers but just qualification of them. You can specify the average increase in correlations and accuracies in the text to give the reader an idea of effect sizes. There is no way to parse if from the text now. I also suggest using actual correlation and classification accuracy values as the main figure (S5 now) rather than the percent changes which will be a more direct report of results.

- Clarify that the functional searchlights are derived from the movie and applied to the localizer experiment in text.

**Have all data underlying the figures and results presented in the manuscript been provided?**

Reviewer #1: Yes

PLOS authors have the option to publish the peer review history of their article (what does this mean?). If published, this will include your full peer review and any attached files.

Reviewer #1: No
---

## [Editor Report · Decision Letter 2]

21 Oct 2020

Dear Dr. Turk-Browne,

We are pleased to inform you that your manuscript 'Searching through functional space reveals distributed visual, auditory, and semantic coding in the human brain' has been provisionally accepted for publication in PLOS Computational Biology.

Best regards,

Saad Jbabdi

Associate Editor

PLOS Computational Biology

Daniele Marinazzo

Deputy Editor

PLOS Computational Biology

---

## [Editor Report · Acceptance letter]

24 Nov 2020

PCOMPBIOL-D-20-00746R2 

Searching through functional space reveals distributed visual, auditory, and semantic coding in the human brain

Dear Dr Turk-Browne,

I am pleased to inform you that your manuscript has been formally accepted for publication in PLOS Computational Biology. Your manuscript is now with our production department and you will be notified of the publication date in due course.

With kind regards,

Nicola Davies
